# Neurodegenerative Disease Associated Pathways in the Brains of Triple Transgenic Alzheimer’s Model Mice Are Reversed Following Two Weeks of Peripheral Administration of Fasudil

**DOI:** 10.3390/ijms241311219

**Published:** 2023-07-07

**Authors:** Richard Killick, Christina Elliott, Elena Ribe, Martin Broadstock, Clive Ballard, Dag Aarsland, Gareth Williams

**Affiliations:** 1Institute of Psychiatry, Psychology and Neuroscience, King’s College London, Denmark Hill, London SE5 8AF, UK; richard.killick@kcl.ac.uk (R.K.); elena.ribe@kcl.ac.uk (E.R.); dag.aarsland@kcl.ac.uk (D.A.); 2College of Medicine and Health, University of Exeter, Exeter EX1 2UL, UK; c.ballard@exeter.ac.uk; 3Faculty of Medical Sciences, Newcastle University, Newcastle upon Tyne NE2 4HH, UK; christina.elliott@newcastle.ac.uk; 4Wolfson CARD, King’s College London, London Bridge, London SE1 1UL, UK; martin.broadstock@kcl.ac.uk

**Keywords:** Alzheimer’s disease, Parkinson’s disease, transcriptional profiling, fasudil

## Abstract

The pan Rho-associated coiled-coil-containing protein kinase (ROCK) inhibitor fasudil acts as a vasodilator and has been used as a medication for post-cerebral stroke for the past 29 years in Japan and China. More recently, based on the involvement of ROCK inhibition in synaptic function, neuronal survival, and processes associated with neuroinflammation, it has been suggested that the drug may be repurposed for neurodegenerative diseases. Indeed, fasudil has demonstrated preclinical efficacy in many neurodegenerative disease models. To facilitate an understanding of the wider biological processes at play due to ROCK inhibition in the context of neurodegeneration, we performed a global gene expression analysis on the brains of Alzheimer’s disease model mice treated with fasudil via peripheral IP injection. We then performed a comparative analysis of the fasudil-driven transcriptional profile with profiles generated from a meta-analysis of multiple neurodegenerative diseases. Our results show that fasudil tends to drive gene expression in a reverse sense to that seen in brains with post-mortem neurodegenerative disease. The results are most striking in terms of pathway enrichment analysis, where pathways perturbed in Alzheimer’s and Parkinson’s diseases are overwhelmingly driven in the opposite direction by fasudil treatment. Thus, our results bolster the repurposing potential of fasudil by demonstrating an anti-neurodegenerative phenotype in a disease context and highlight the potential of in vivo transcriptional profiling of drug activity.

## 1. Introduction

Drug repurposing is the application of an existing therapeutic, with a known safety profile and prescription data, to a new condition. This strategy is particularly relevant to neurodegenerative conditions that have so far proven refractory to traditional target-based novel entity development [1]. Repurposing offers a quick route to the clinic following the identification of a novel target for the given disease, where one can adopt an off-the-shelf approach benefiting from the extensive drug development data and post-approval patient histories. In this context, inhibition of the ROCKs, a target for vasodilation post-ischemia for which there is a clinically approved drug, has emerged as a possible strategy in the treatment of neurodegenerative diseases (NDDs).

Fasudil (HA-1077), a vasodilator, was developed by the Hidaka group during the 80s [2] and initially thought to be a Ca^2+^ channel antagonist [3,4] with particularly potent effects on the vasculature supplying the brain [2,4]. However, following the identification of ROCK, it very soon became apparent that the drug acts on blood vessels through the antagonism of both ROCK1 and ROCK2, showing similar IC50s towards each, 730 and 720 nM, for ROCK1 and ROCK2, respectively [2]. Fasudil gained clinical approval for the treatment of cerebral vasospasm following subarachnoid haemorrhage (SAH) in Japan in 1994 and then in China [5]. Interestingly, fasudil was observed to also improve neuronal function in animal models of SAH [6]. This led Kamei et al. to test fasudil in two human subjects with cerebrovascular dementia with wandering [7]; the authors reported that when given orally at 30 or 60 mg a day for 8 weeks, fasudil prevented wandering and improved memory. Further, positive effects on cognitive performance and measures of daily living were reported for Fasudil in a small clinical trial of the mild cognitively impaired [8].

A genomic association study implicated the human gene *WW1C*/*KIBRA* in memory function [9]. The observation that the *KIBRA* is negatively regulated by ROCK led Huentalman et al. [10] to test and show that ROCK inhibition, using hydroxyfasudil (the active major metabolite of fasudil), can potentiate memory and even reduce age-dependent memory impairment in rats. This was one of the first reports on the preclinical use of fasudil in the West and gained the attention of the popular media, generating headlines such as: “Drug found that could reduce risk of Alzheimer’s disease”, which appeared in the online publication ScienceDaily [11].

In this context, RhoA/ROCK signalling activity is intimately involved in synapse remodelling [12] and retraction [13] and inflammatory responses in many tissues, including the brain [14]. Like Alzheimer’s disease (AD), many neurodegenerative diseases (NDDs) also feature synapse loss and neuroinflammation [15]. As such, fasudil, and a number of other small molecule antagonists of ROCK, have been used in numerous preclinical studies in animal models across many different NDDs [16], with fasudil repeatedly being found to offer benefits in such studies [17,18,19,20].

To investigate the mechanism of fasudil action in the context of NDD pathology further, we performed a global transcriptional analysis of brain tissue from fasudil-treated 3xTG-AD mice. This model captures the two major AD neuropathologies, senile plaques (6 months) and neurofibrillary tangles (12 months), and with cognitive impairment at just 4 months [21]. After peripheral administration of fasudil via IP injection regimen over two weeks, we found significant gene expression changes in the brains of the treated mice. By comparing our expression profile with data generated from publicly available post-mortem NDD sample data, including AD, Parkinson’s disease (PD), and Huntington’s disease (HD), we show that fasudil tends to drive gene expression in the opposite sense to that in NDD. This effect is more pronounced in the genes down-regulated in NDD and emerges most clearly through a pathway enrichment analysis. Our results highlight the potential of transcription profiling to reveal the therapeutic potential of candidate therapeutics in the context of NDD. Specifically, we show how expression profiling of fasudil activity at the site of disease pathology reveals its disease-ameliorating potential through its driving gene expression in an opposite sense to that found in disease. 

## 2. Results

### 2.1. Expression Changes in the Brains of 3xTg-AD Mice Post-Fasudil Treatment

To assess the global transcriptional effects on the brains of AD model mice post-fasudil treatment, we performed a SOM analysis, as described in the Methods section below. A SOM consists of a matrix of weights, each capturing distinct gene expression variation in the experiment. The amount of variation explained by fasudil treatment can be gauged by regressing the weights against treatment status. We find that the SOM segregates into islands representing genes whose expression is up/down-regulated by fasudil; see Figure 1. To assess the significance of the weight correlations with treatment status, we generated 10,000 random permutations of treatment assignment and observed that 22 permutations had maximal weight correlation equal to or in excess of that for the actual treatment assignment. This implies a probability of 0.00022 for the null hypothesis of there being no transcriptional effect of fasudil treatment. A similar analysis with sex points to a more modest expression change; see Figure 1. We further generated a SOM with the effects of sex subtracted with a linear model fit across the samples. Here, the fasudil effects are more striking; see Figure 1. 

Performing a differential expression analysis, we found that the expression changes driven by fasudil in the brains of the 3xTg-AD mice were relatively modest; see Appendix A. Restricting the analysis to these genes will only furnish a limited analysis of biological changes driven by fasudil. We, therefore, decided to base our analysis on the global expression profile defined as the ranked Z scores corresponding to the gene changes between control and treatment groups. As described in the Methods section, we defined a fasudil profile by fitting expression levels to treatment category and sex. The profile then consists of 15,699 genes, of which 478 and 414 are up- and down-regulated at two standard deviations away from the mean, respectively; see Appendix A. 

### 2.2. Fasudil-Driven Gene Expression and Neurodegenerative Disease

To obtain an idea regarding the biological systems that are being perturbed by fasudil treatment, we performed a pathway analysis along the lines described in detail in our recent paper [22]. Here, the full set of genes in the expression profiles was ranked according to the Z scores of the expression changes across the samples, and a pathway enrichment analysis was performed. 

We found an up-regulation of NGF signalling genes (*p* < 1.78 × 10^−4^) as well as pathways down-regulated in AD [23], PD (KEGG)(*p* < 8.59 × 10^−4^), and HD (KEGG) (*p* < 3.59 × 10^−3^). Oxidative phosphorylation has been shown to be down-regulated in AD [24,25,26,27], and we found that fasudil up-regulates metabolic pathways: oxidative phosphorylation (*p* < 8.17 × 10^−4^), Voxphos (*p* < 9.93 × 10^−3^), mitochondria (*p* < 2.93 × 10^−3^), and respiratory electron transport (*p* < 3.52 × 10^−4^). Wnt signalling is reported to be down-regulated in AD [28,29], and we found that fasudil drives this pathway up (*p* < 5.29 × 10^−3^). 

The observation that pathways associated with AD and other neurodegenerative diseases are significantly regulated led us to investigate a possible overlap with transcriptional changes in the AD brain. To this end, we generated pathways for a composite AD profile described in Reference [30]. At the level of perturbed genes, we can see that there is indeed a significant anti-correlation with AD and other NDD profiles (PD and HD), as shown in the top-ranked gene list and the contingency table at the |Z| > 2 threshold in Figure 2 and Table 1. The reversal is particularly evident with those genes down-regulated in AD. Of the shared genes at the Z > 2 level, 20 of the top 25 most down-regulated genes in AD, based on Z score rank, were up-regulated by fasudil treatment. The picture is less clear for the genes up-regulated in AD, with 14 in the top 25 down-regulated by fasudil treatment.

A pathway enrichment analysis provides insight into the underlying biology and can facilitate a comparison between expression profiles that is less sensitive to the noise component in gene expression. In this context, it is interesting to observe that the contrast between NDD and fasudil is more clearly delimited through a comparative pathway analysis. An overall picture of the pathways regulated in NDD and by fasudil is given in Figure 3, and the contingency table plus statistics are in Table 2. In the pathway set down-regulated in AD, those that are regulated by fasudil are overwhelmingly driven upwards (all in the top 25). The top 25 up-regulated pathways seen in AD 19 are down-regulated by fasudil treatment.

### 2.3. Gene Expression Changes in the 3xTg-AD Model in Relation to Those Seen in AD

The 3xTg-AD mouse model harbours human AD risk variants in APP, presenilin-1, and tau [21]. The model exhibits both amyloid plaques and tau tangles [17,21] as well as gliosis [31] and presents relatively early cognitive deficits [32]. The 3xTg-AD mice have been extensively investigated at the transcriptional level and shown to recapitulate neuronal inflammation [33,34], the dysregulation of genes involved in learning and memory [35], and psychiatric disorders [36]. With a view to testing the robustness of the model’s relationship to AD at the global transcriptional level, we performed a meta-analysis of publicly available data on the gene expression changes in 3xTg mouse brains relative to wild-type animals across multiple time points; see Methods section. We generated 12 distinct profiles and found that they are largely consistent, with an average correlation score of 6.88 standard deviations from the null. To make a connection to human disease, we combined the separate 3xTg-AD profiles into a summary Z score profile; see Appendix A. We found that the expression changes in the composite 3xTg-AD are significantly correlated with the composite AD profile; see Appendix A. In particular, at the |Z|≥2 level, there are 1571 shared genes, of which 1114 have expression changes in the same sense with an associated Fisher exact test of 1.49E-40. Performing a regression analysis, we find a Pearson correlation coefficient of 0.33 with an associated Z score of 13.45. Also, consistent with our analysis above, we find that the expression changes driven by fasudil in the 3xTg-AD mouse are significantly anti-correlated with those in 3xTg-AD relative to wild-type animals; see Appendix A. In particular, of the 157 pathways that are perturbed in the 3xTg-AD mice versus wild-type mice and the fasudil-treated 3xTg-AD mice, all but 9 are perturbed in the reverse sense; see Appendix A. Again, in terms of gene-by-gene comparison, the anti-correlation is less pronounced, with 144 out of 198 genes being perturbed in a reverse sense, Appendix A.

### 2.4. Fasudil-Driven Transcription In Vitro

We performed pathway enrichment analyses on all the CMAP compounds as described in the Methods section. The pathway profiles of the CMAP drugs enabled us to perform an extensive comparison of the disease pathway profiles. The relationship between disease and drug candidate can be represented in terms of a distance measure, $d=\frac{1}{2}\left(1+c\right)$, where −1≤c≤1 is the correlation. So, drug candidates are hypothesised to be close to the disease; see Methods. We find that in terms of a pathway profile comparison, fasudil emerges as a candidate in the top 9% of anti-correlating profiles for AD; see Figure 4. In contrast, a similar analysis based on gene-wise comparisons fails to show any anti-correlation between the CMAP fasudil profile and that of AD; see Figure 4. 

However, the relatively weak anti-correlation of fasudil with NDD in the pathway landscape speaks to the importance of assessing compound activity in a disease-relevant context, as the therapeutic potential of fasudil would have been missed with an analysis of cell line-based compound expression profile database searches, such as those provided by CMAP and LINCS [37].

## 3. Discussion

Several groups, including our own, have reported observations that demonstrate that the administration of the pan ROCK inhibitor, fasudil, to animal models of AD is beneficial [17,18,38,39,40]. Fasudil has also shown beneficial effects on memory and wandering in people suffering from cerebrovascular dementia [7], and Yan et al. [8] have reported positive effects on cognitive performance and measures of daily living in a small clinical trial in a cohort of the mild cognitively impaired.

Here, for the first time, we examined the transcriptomic effects of fasudil in the frontal cortex of a rodent AD model, the triple transgenic mouse, following peripheral administration of fasudil by IP injection twice daily at 10 mg/kg for two weeks.

An initial global analysis of the microarray gene expression data through a self-organising map reveals a significant effect of fasudil in the frontal cortices of treated animals. With a view to uncovering the potential AD-reversing effects of fasudil, we compared its differential expression profile with composite expression profiles derived from various neurodegenerative diseases. Interestingly, we found that there is a significant anti-correlation between the fasudil profile and that of AD, PD, and HD. This is particularly evident in the genes down-regulated in the neurodegenerative conditions. For example, amongst the genes whose expression is significant, |Z|≥2, altered in both the summary AD and fasudil profiles, 80% of the most down-regulated genes in AD, according to the Z score, are up-regulated by fasudil. Differential expression analysis is complicated by the contribution of noise and the large gene-to-sample ratio. One way of smoothing noise is to perform a pathway enrichment analysis, as this is an analysis based on sets of genes where we expect noise effects to cancel. To this end, we performed a pathway enrichment analysis on the fasudil-driven expression profile and the composite neurodegenerative disease profiles. Comparing the pathways significantly regulated in disease and drug treatment, we found a clear reversal, particularly in the pathways down-regulated in disease. Here, of the top 25 down-regulated pathways in AD that are regulated by fasudil, all are driven upwards by the drug. 

It is worthwhile to point out here that in vivo fasudil will be rapidly converted to its major metabolite, hydroxyfasudil, which has a much longer half-life and very similar IC50 values towards ROCK1 and ROCK2 as fasudil [41].

The potential of pathway enrichment analysis in drug repurposing is highlighted by our analysis of deposited data on drug-driven expression profiles in cell lines. In particular, our composite neurodegenerative disease pathway sets show a relatively high anti-correlation with those of fasudil in the CMAP database. For example, fasudil is in the top 9% of hits for AD ordered by anti-correlation. This is in contrast with a gene-wise analysis, where there is no significant anti-correlation between the neurodegenerative profiles and fasudil in the same CMAP database. However, the pathway analysis at the in vitro level reveals a much weaker connection between fasudil and disease than seen in vivo. It will be interesting to see whether the anti-neurodegenerative phenotype of fasudil is revealed in the in vivo context because we are looking at responses in the appropriate tissue, or whether these disease-reversing effects will only manifest in a disease model.

Further to our own work with fasudil, we found that it protects synapses in the face of synaptotoxic amounts of oligomeric forms of Aβ_1-42_ [17,18], that it protects memory following intracerebroventricular injection of oligomeric Aβ, reduces the neuronal production of endogenous Aβ species 1-38, 1-40, and 1-42, and reduces levels of both soluble Aβ and senile plaque-like deposits in the 3xTg-AD mouse brain after just two weeks of peripheral administration [17]. In further studies we have now completed and are in the process of publishing, we found that fasudil also reduces tau pathology, including a reduction in phospho-T181 tau immunoreactivity, in brains of the TgF344-AD rat (RK, unpublished observation), a model of AD which develops tau pathology without the addition of a tau transgene as is present in the 3xTg-TD mouse, and, importantly, in neurospheroids generated from human iPSC-derived neurons obtained from AD patients (RK, unpublished observation). In addition, fasudil also reduces both brain microglial and astrocytic immunoreactivity and, by MRI-based measures, prevents brain atrophy and neuronal loss in the Tg-F344-AD rat (RK, unpublished observation).

This novel analysis of the transcriptional effects of peripherally delivered fasudil on brain tissue not only demonstrates that the drug is brain permeable and strengthens our contention that it will be of benefit for treating AD, but also supports the contentions of others that fasudil could also be of benefit for treating PD and other synucleinopathies [42] and HD [43,44]. The data we present here, supporting the use of fasudil for AD, PD, and HD, is further supported by an earlier proteomic-based analysis of fasudil which pinpointed the same three neurodegenerative diseases [45].

## 4. Materials and Methods

### 4.1. Drug Treatment

At 18 months of age, 3xTg-AD mice were administered 10 mg/kg fasudil (*n* = 7, 3 female and 4 male) or PBS vehicle (*n* = 10, 6 female and 4 male) twice daily, IP, for two weeks. Mice were sacrificed, and brains were removed, hemisected, and fixed or snap-frozen, as previously described [17]. Rostral cortices were subsequently collected from the snap-frozen hemibrains after partial thawing on ice, and total RNA was extracted by homogenisation directly in TRIzol reagent and isopropanol precipitation, then further purified using RNeasy ^®^ mini kit (Ref: 74104, Qiagen, US).

### 4.2. Microarray Analysis

RNA was converted to cRNA and hybridised to full genome Affymetrix Mouse Gene 2.0 ST microarrays, processed, and imaged according to the manufacturer’s instructions. The CEL files were processed in the R environment using the Bioconductor oligo package to give RMA normalised expression values [46]. 

The self-organizing map (SOM) analysis [47] was performed on the expression data after it was normalised to N(0, 1) across samples. Random weights were defined on an 10 × 10 square lattice and mapping optimization was performed using a Euclidean distance metric. The iteration is as follows:$${\vec{w}}_{ij}={\vec{w}}_{ij}+\delta \left(\vec{g}-{\vec{w}}_{ij}\right)e^{-{\left(i^{2}+j^{2}-\left(i_{0}^{2}+j_{0}^{2}\right)\right)}^{2}/\sigma^{2}}$$
where ${\vec{w}}_{ij}$ are the SOM weights and the vector refers to the space of samples, $\vec{g}$ is expression level of the given gene across the samples, and the weight closest to $\vec{g}$ is at $\left(i_{0},j_{0}\right)$. The step size δ is set to 0.01 and reduced by a factor of 0.99 upon each iteration. The weight neighbourhood extends over the Gaussian standard deviation σ, and this is initialised at 2 and reduced by a factor of 0.99 upon each iteration.

The weights were then regressed against treatment status or animal sex. A further SOM was performed on the residual expression data with sex variance subtracted.

Expression profiles were defined as the linear fit Z scores for treatment versus control samples and the genes mapped to the HUGO Gene Nomenclature Committee (HGNC) names (www.genenames.org). The fasudil samples comprised both male and female mice, and the Z scores were generated based on a linear model with sex as a covariate. The probes were mapped to genes with the maximal magnitude Z score selected in cases of alternative probes; see Appendix A for the full profile. 

### 4.3. Meta-Analysis Transcription Profiles

The AD profile was based on the composite Z scores for 21 profiles from 13 series corresponding to post-mortem brain samples that were publicly available on the NCBI GEO data repository [48], as described by us in Reference [23]. Composite scores were defined to correspond to the average Z score across the profiles; the average is obtained by dividing the sum by the square root of the number of profiles. Missing values are set to zero.

In particular, the expression series and samples are as follows: NCBI GEO accession: GSE84422 (frontal pole 24 v 15 and para hippocampal gyrus 23 v 15) [49], GSE37263 (neocortex 8 v 8) [50], GSE36980 (hippocampus 7 v 10, frontal cortex 15 v 18, and temporal cortex 10 v 19) [36], GSE39420 (posterior cingulate 7 v 7) [51], GSE1297 (hippocampus severe AD 7 v 9 and moderate AD 8 v 9) [52], GSE29378 [53], GSE48350 (entorhinal cortex 15 v 15) [54], GSE15222 (hippocampus severe AD 7 v 9 and moderate AD 8 v 9) [55], GSE26972 (entorhinal cortex 3 v 3) [56], GSE37264 (neocortex 8 v 8) [57], GSE28146 (severe AD 7 v 8 and moderate AD 7 v 8) [58], GSE5281 (medial temporal gyrus 16 v 12 and entorhinal cortex 10 v 13) [59], and GSE13214 (frontal cortex 20 v 18) [60].

The PD composite profile was defined in a similar way to the AD profile and was based on the 12 GEO expression series: GSE8397 (medial substantia nigra 15 v 6 GPL96 plus GPL97, lateral substatia nigra 9 v 5 GPL96 plus GPL97, and superior frontal gyrus 5 v 3 GPL96 plus GPL97) [61], GSE46036 (substantia nigra alphasynuclein 8xBRAAK5/6 v 8xBRAAK0, 7xBRAAK3/4 v 8xBRAAK0, and 5xBRAAK1/2 v 8xBRAAK0) [62], GSE20291 (putamen 15 v 20) [63], GSE20164 (substantia nigra 6 v 5) [64], GSE49036 (substantia nigra alphasynuclein 8xBRAAK5/6 v 8xBRAAK0, 7xBRAAK3/4 v 8xBRAAK0, and 5xBRAAK1/2 v 8xBRAAK0) [65], GSE7621 (substantia nigra 16 v 9) [66], GSE19587 (inferior olivary nucleus 6 v 5 and vagus dorsal motor nucleus 6 v 5) [67], GSE34516 (locus coeruleus 4 v 4 and LRRK2 2 v 4) [68], GSE20314 (cerebellum 4 v 4) [64], GSE43490 (substantia nigra 3 v 6) [69], GSE54282 (cortex 5 v 5) [70], and GSE2437 8 (substantia nigra dopamine neurons 8 v 9) [64]. This resulted in a composite from 21 distinct PD profiles. The HD profile was based on the GEO series GSE3790 [71]. The HD series covers caudate nucleus covering frontal cortex and cerebellum, comprising 114 samples of varying HD severity and 87 controls. Here, the expression levels were fit with a linear model with age, sex, and brain region as covariates. The profiles are given in Appendix A.

The expression changes in the triple transgenic mouse model of AD were derived from three RNAseq series (GSE168428 (3 3xTG v 3 WT at 2 months, 6 3xTG v 6 WT at 7/8 months, and 6 3xTG v 6 WT at 11/14 months) [33], GSE161904 (4 3xTG v 4 WT at 3 months, 8 months, and 15 months) [35], and GSE189693 (4 3xTG v 4 WT at 10 months) [72]) and five microarray series (GSE35210 (3 3xTG v 3 WT at 12 months) [73], GSE36981 (3 3xTG v 3 WT at 14 months) [36], GSE60460 (4 3xTG v 4 WT at 7 months), GSE60911 (5 3xTG v 5 WT at 20 months) [74], and GSE92926 (3 3xTG v 3 WT at 12 months) [34]) resulting in 12 transcriptional profiles that were combined into a single composite Z score profile; see Appendix A.

### 4.4. Pathway Analysis

Pathway gene sets were obtained from the Broad Molecular Signature database [75] and correspond to the canonical pathway C2 collection of 2450 sets of genes. The significance of a pathway being perturbed in an expression profile is determined by a non-parametric Kolmogorov–Smirnov type test on the deviation of the cumulative distribution of pathway genes on the ranked expression profile relative to the null hypothesis linear distribution. Specifically, the pathway enrichment score is defined as the Z score corresponding to the ratio of the sum of the maximal deviation above and below the null to the expected standard deviation [22]. Expression profiles are ranked based on the gene’s Z score of expression change over the given sample category assignments.

### 4.5. CMAP Pathway Enrichment Profiles

The Connectivity MAP 2.0 [76] comprises gene expression change data for the activity of 1309 drug-like compounds on cancer cell lines. The data were downloaded in the form of ranked probes for the given drug treatments relative to the plate controls. These data were mapped to average probe ranks for each compound resulting in 1309 profiles. Pathway profiles, defined as the lists of significantly enriched pathways for the expression profiles, were then generated for each compound. 

To compare the profiles of perturbed genes/pathways in neurodegenerative disease with those in CMAP, we defined a correlation score as $c=\frac{UU+DD-UD-DU}{UU+DD+UD+DU}$, with U and D being the number of shared pathways/genes regulated upwards and downwards, respectively. To represent the relationship of the disease profile to the drug profiles on a plane, the distances between drugs were defined as given $d=\frac{1}{2}\left(1-c\right)$ and the distances of a drug to the disease profile as $d=\frac{1}{2}\left(1+c\right)$, such that proximity now indicates anti-correlation. Points are mapped to the plane with the target at the centre, in black, and distances to the target are preserved. Angular separations of the compounds are optimised through steepest descent iteration where the function to be minimised is
$$\Phi =\underset{i>j}{\sum }{\left(d_{ij}-\left|{\vec{r}}_{i}-{\vec{r}}_{j}\right|\right)}^{2}$$
where ${\vec{r}}_{i}$ is a two-dimensional vector representing the $i^{\mathrm{th}}$ profile with the disease profile, i=0, at the origin, ${\vec{r}}_{0}=\vec{0}$. Φ is minimised as a function of the compound profile angles $\theta_{i}$, with ${\vec{r}}_{i}=r_{i}cos\theta_{i}+r_{i}sin\theta_{i}$ and $r_{i}=d_{i0}$. The iteration starts with a random assignment of angles and proceeds as follows:$$\theta_{i}^{k+1}=\theta_{i}^{k}-\gamma^{k}\partial_{i}\Phi^{k}$$
$$\gamma^{k}=\frac{\sum_{i}\left(\theta_{i}^{k}-\theta_{i}^{k-1}\right)\left(\partial_{i}\Phi^{k}-\partial_{i}\Phi^{k-1}\right)}{\sum_{i}\left(\partial_{i}\Phi^{k}-\partial_{i}\Phi^{k-1}\right)\left(\partial_{i}\Phi^{k}-\partial_{i}\Phi^{k-1}\right)}$$

## Figures and Tables

**Figure 1 ijms-24-11219-f001:**
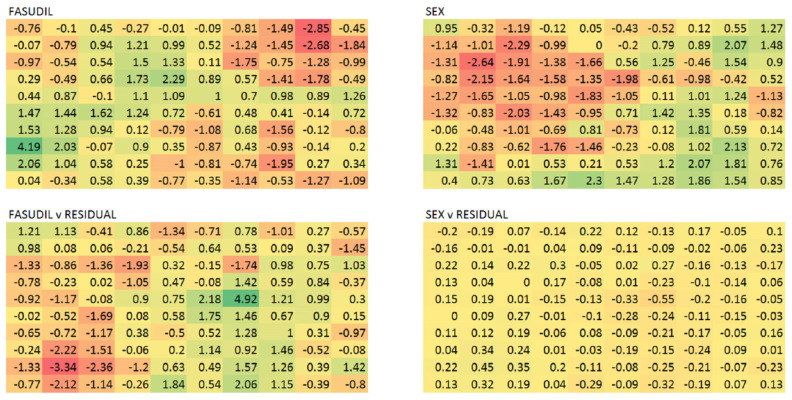
Self-organising maps of the transcriptional data reveal the global effects of fasudil treatment. In the top left, the SOM weights are regressed against treatment status, and the corresponding Z scores are shown, coloured to represent the scores. It is clear that there are islands of positive and negative correlation corresponding to genes that are up- and down-regulated, respectively. On the top right, the weights are correlated with the mouse sex and show a significant sex effect that is, however, muted relative to that of fasudil. The expression data with the variance explained by sex subtracted out leaves a residual expression profile for which the SOM is shown below. As expected, sex no longer shows any significant correlation, bottom right, with the weights, and interestingly, the fasudil correlations are stronger, bottom left.

**Figure 2 ijms-24-11219-f002:**
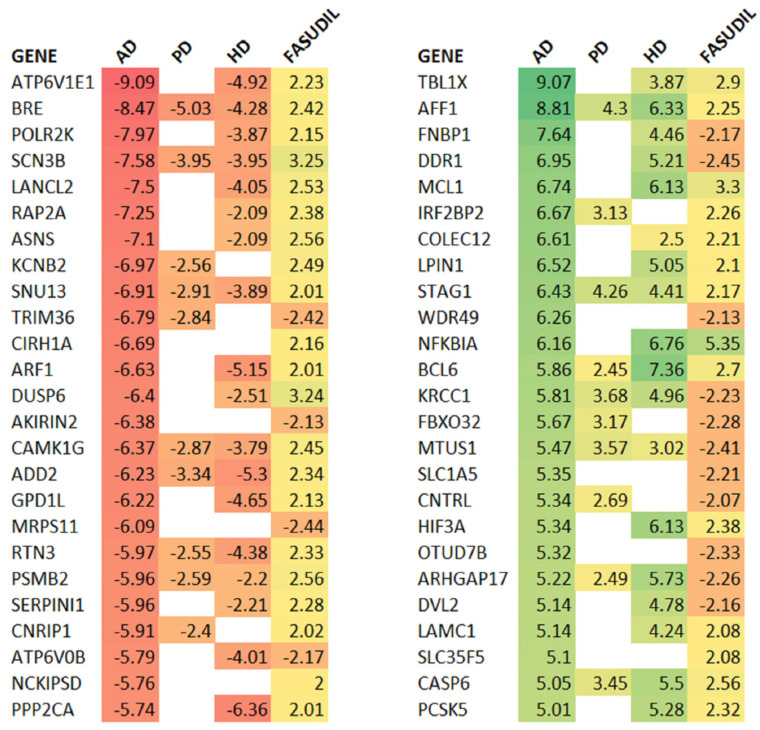
Fasudil-driven gene expression changes tend to be the reverse of those in multiple neurodegenerative conditions. The significant gene expression Z scores (|Z|>2) are shown for AD, PD, and HD together with those for the fasudil treatment; non-significant values are left blank, and entries are coloured according to the scores. The most down-regulated genes in AD are shown on the left, and the most up-regulated on the right. As expected, there is a high degree of consistency across the other neurodegenerative conditions, PD and HD. Fasudil shows an up-regulation of all but four genes of those most down-regulated in AD. The reversal is less significant for those genes up-regulated in AD, with only 14 out of 25 driven down with fasudil.

**Figure 3 ijms-24-11219-f003:**
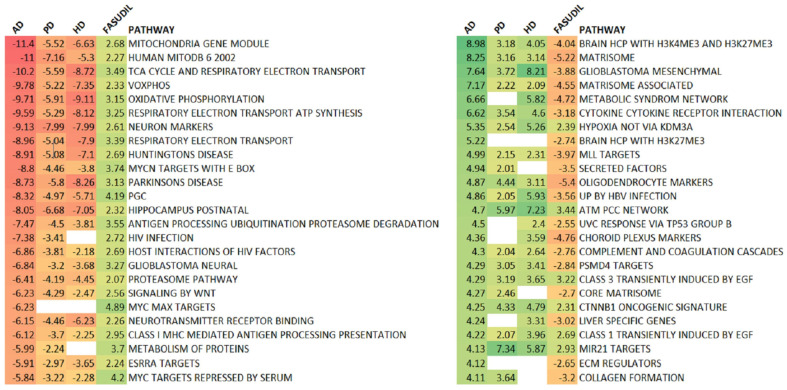
Fasudil regulates pathways in an opposite sense to that seen in multiple neurodegenerative conditions. The significant pathway enrichment Z scores (|Z|>2) are shown for AD, PD, and HD together with those for the fasudil treatment; non-significant enrichment scores are left blank with entries coloured according to the scores. The most down-regulated pathways in AD are shown on the left, and the most up-regulated on the right. As expected, there is a high degree of consistency across the other neurodegenerative conditions, PD and HD. Fasudil shows an up-regulation of all the pathways most down-regulated in AD and a down-regulation of 76% of the most up-regulated pathways in AD.

**Figure 4 ijms-24-11219-f004:**
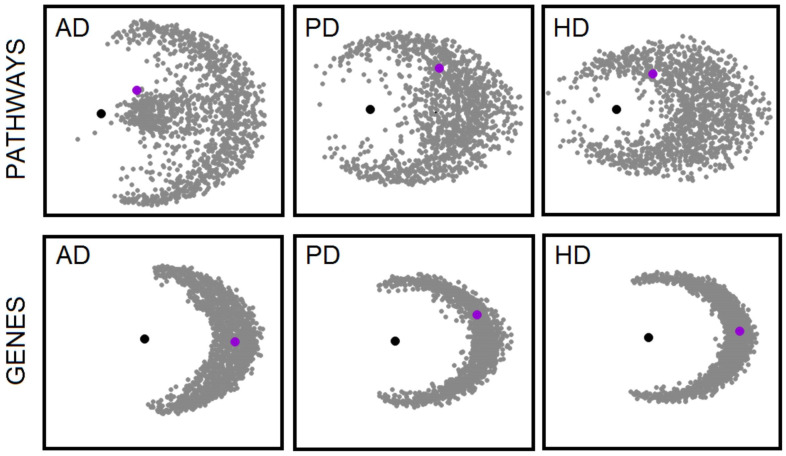
Pathway and gene-based analyses of the CMAP drug-driven transcription profiles in cancer cell lines in relation to target profiles from AD, PD, and HD. The drugs and target profiles are shown as points on a plane with distances between drugs given by $\frac{1}{2}\left(1-c\right)$, where the correlation is $c=\frac{UU+DD-UD-DU}{UU+DD+UD+DU}$, with U and D the number of shared pathways/genes regulated upwards and downwards, respectively. The distance to the target profile is $\frac{1}{2}\left(1+c\right)$, such that proximity now indicates anti-correlation. Points are mapped to the plane with the target at the centre, in black, and distances to the target are preserved. Angular separations of the compounds are optimised through steepest descent iteration. The compounds are scattered in a more diffuse pattern for the pathway-based analysis. Interestingly, fasudil, shown in violet, emerges as a repurposing candidate only in the pathway-based analysis. In particular, the contingency table for AD is $\left[\begin{array}{cc} 12 & 20\\ 41 & 11 \end{array}\right]$ p<0.00017, that for PD is $\left[\begin{array}{cc} 17 & 13\\ 24 & 6 \end{array}\right]$ p<0.047, and that for HD is $\left[\begin{array}{cc} 20 & 12\\ 36 & 6 \end{array}\right]$ p<0.016.

**Table 1 ijms-24-11219-t001:** The correlations of the fasudil transcription profile with neurodegenerative disease profiles. The contingency tables for regulated genes at the |Z|>2 level are shown at the top for AD, PD, and HD. The corresponding correlation scores, measured by $\frac{UU+DD-\left(UD+DU\right)}{UU+DD+UD+DU}$, where the U and D stand for up- and down-regulated genes, are shown below. The significance of one-sided Fisher exact test is shown in brackets. The fasudil profile significantly anti-correlates with the AD, PD, and HD profiles. The neurodegenerative disease profiles all show strong positive correlations with each other.

	AD		PD			HD		
**FASUDIL**	−0.28 (2.87 × 10^−11^)		−0.30 (4.62 × 10^−8^)			−0.13 (1.43 × 10^−5^)		
**AD**			0.92 (5.00 × 10^−324^)			0.86 (5.00 × 10^−324^)		
**PD**						0.81 (2.14 × 10^−286^)		
**AD**	**U**	**D**	**PD**	**U**	**D**	**HD**	**U**	**D**
**U**	107	133	**U**	48	52	**U**	121	94
**D**	109	29	**D**	45	4	**D**	98	26

**Table 2 ijms-24-11219-t002:** The correlations of the fasudil treatment-regulated pathway set with those perturbed in neurodegenerative disease profiles. The contingency tables for regulated pathways at the |Z|>2 level are shown at the top for AD, PD, and HD. The corresponding correlation scores, measured by $\frac{UU+DD-\left(UD+DU\right)}{UU+DD+UD+DU}$, where the U and D.

	AD		PD			HD		
**FASUDIL**	−0.60 (7.10 × 10^−28^)		−0.28 (3.74 × 10^−5^)			−0.11 (4.78 × 10^−5^)		
**AD**			0.98 (5.47 × 10^−105^)			0.96 (2.22 × 10^−122^)		
**PD**						0.99 (3.42 × 10^−96^)		
**AD**	**U**	**D**	**PD**	**U**	**D**	**HD**	**U**	**D**
**U**	45	104	**U**	45	66	**U**	73	52
**D**	74	0	**D**	20	3	**D**	46	6

## Data Availability

Not applicable.

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
