# Peer review of "Neurodegenerative Disease Associated Pathways in the Brains of Triple Transgenic Alzheimer’s Model Mice Are Reversed Following Two Weeks of Peripheral Administration of Fasudil"

_ijms, 2023, doi:10.3390/ijms241311219_

Round 1

Reviewer 1 Report

Concerns and Suggestions Regarding the Study on ROCK Inhibitor "Fasudil" in Neurodegenerative Disorders

Dear Authors

I hope this email finds you well. I have recently reviewed your study on the effects of the ROCK inhibitor "fasudil" in neurodegenerative disorders. While I appreciate the research you have conducted, I have identified several concerns that I believe could enhance the study's clarity and impact. I would like to offer some suggestions to address these concerns and improve the overall quality of your work.

1.     Writing Clarity: The manuscript could benefit from improved writing and overall clarity. I recommend ensuring that figure numbers and figure legends are clearly indicated throughout the text. This will facilitate easy reference and comprehension for readers. I can see Figure 1 written in text, but I could not find figure with this name. Please also add figure ligands.

2.     Color Coding: Including color coding in your figures can be immensely helpful in visually representing different data or groups. I suggest explaining the color-coding scheme used in your figures, providing readers with a clear understanding of the meaning behind the colors and their relevance to different conditions or variables.

3.   Unclear table: It appears that there are issues with numbers printed on page 6 and 7 of the tables. I recommend carefully reviewing and correct them.

4.     Gene Expression Patterns: It is important to address the observation that, in most cases, gene expression goes opposite to the disease conditions following fasudil treatment. The expression of genes changes more than 2-folds in opposite direction, which is also not normal and can leads to some abnormalities. Which should be addressed.

5.   Brain histopathological analysis and behavior study can reveal the improve disease conditions after the treatment.

6. Adding one more control, where wild-type animals should be treated with the fasudil and expression of genes reported in the neurodegenerative disorders should be checked. This will clear that the fasudil actually can upregulate those genes or it is only in the neurodegenerative disorder condition.

By addressing these concerns and incorporating the suggested improvements, your study will become more coherent, transparent, and scientifically robust. I believe these enhancements will contribute significantly to the impact and understanding of your research findings.

Author Response

We have addressed the comments of the reviewers, see individual replied to queries in italics below. There appears to have been a problem with the conversion of the Word doc file to pdf format and consequently Figure legends were lost and some line numbers got entangled with Table data. To address this problem, we have now uploaded a locally generated pdf file.

  1. Writing Clarity: The manuscript could benefit from improved writing and overall clarity. I recommend ensuring that figure numbers and figure legends are clearly indicated throughout the text. This will facilitate easy reference and comprehension for readers. I can see Figure 1 written in text, but I could not find figure with this name. Please also add figure ligands.

Unfortunately, the MS got somewhat corrupted upon pdf generation. We have now loaded a pdf version of the paper. Hopefully, this will now clarify our work.

  1. Color Coding: Including color coding in your figures can be immensely helpful in visually representing different data or groups. I suggest explaining the color-coding scheme used in your figures, providing readers with a clear understanding of the meaning behind the colors and their relevance to different conditions or variables.

The colour coding is explained in the Figure legends that were missing in the original compilation, see comment above. In brief, the colour scale is defined by the values in the cells. Colours are given purely as visual aids supplementing the numerical data in the tables.

  1. Unclear table: It appears that there are issues with numbers printed on page 6 and 7 of the tables. I recommend carefully reviewing and correct them.

This has been dealt with and is again a consequence of the MS corruption upon pdf generation.

  1. Gene Expression Patterns: It is important to address the observation that, in most cases, gene expression goes opposite to the disease conditions following fasudil treatment. The expression of genes changes more than 2-folds in opposite direction, which is also not normal and can leads to some abnormalities. Which should be addressed.

The motivation for our investigation of the transcriptional effects of fasudil in the brains of Alzheimer’s disease (AD) model mice was to gain a mechanistic insight into the beneficial effects of the drug in the context of neurodegenerative disease. The hypothesis behind transcription-based drug repurposing holds that beneficial drugs will tend to drive gene expression in the reverse sense to that seen in the given disease. Our present results show that fasudil indeed emerges as a transcription-based repurposing candidate as it reverses gene expression changes in both the AD model mouse brains and those seen in post-mortem AD brains.

  1. Brain histopathological analysis and behavior study can reveal the improve disease conditions after the treatment.

We agree and this work has indeed been done with fasudil.  We have referenced previous data on the beneficial effects of fasudil in the context of AD. The present work seeks to show that these beneficial effects are encoded in gene expression changes driven by the drug in brain tissue.

  1. Adding one more control, where wild-type animals should be treated with the fasudil and expression of genes reported in the neurodegenerative disorders should be checked. This will clear that the fasudil actually can upregulate those genes or it is only in the neurodegenerative disorder condition.

The motivation behind this investigation was to see to what extent the beneficial activities of fasudil in relation to AD were manifest at the level of transcription and this is why we looked at the transcriptional effects of fasudil in the AD model mouse. Our conclusion is in broad agreement with our hypothesis that fasudil will drive transcription in the reverse sense to that seen in the disease state. As far as looking at the transcriptional activities of fasudil in the non-AD context, we have presented the cell line data and shown that the repurposing potential of fasudil, encoded by gene expression changes, is much diminished and only emerges at the level of a pathway analysis. An intermediate analysis would be to look at fasudil activities in vivo in wild-type animals. This would be of general interest regarding the extension of cell line data, but it is difficult to see how this data could affect our conclusions. 

Reviewer 2 Report

This is an interesting report. The authors carried out a comparative analysis on the brains of AD model mice treated with fasudil  and other neurodegenerative diseseases (IP injection). It came out that fasudil "tends" to drive gene expression in the opposite sense to that observed in AD and PD.

Some ninnor points:

1, Abstract: Not Re-sults but results.

2. In the subchapter Results: not mehodes above, but methods below.

3. Try to edit tables. They are oddly printed (at least in the text version the reviewer had)

Author Response

This is an interesting report. The authors carried out a comparative analysis on the brains of AD model mice treated with fasudil  and other neurodegenerative diseseases (IP injection). It came out that fasudil "tends" to drive gene expression in the opposite sense to that observed in AD and PD.

Some ninnor points:

1, Abstract: Not Re-sults but results.

We thank the referee for pointing out this error and have corrected it.

  1. In the subchapter Results: not mehodes above, but methods below.

We thank the referee for pointing out this error and have corrected it.

  1. Try to edit tables. They are oddly printed (at least in the text version the reviewer had)

Unfortunately, the MS got somewhat corrupted upon pdf generation. We have now loaded a pdf version of the paper. Hopefully, this will now clarify our work.

Reviewer 3 Report

The manuscript titled “Neurodegenerative disease associated pathways in the brains of triple transgenic Alzheimer’s model mice are reversed following two weeks peripheral administration of fasudil”, shows an analysis of the transcriptional effects of peripherally administered fasudil in brain tissue, the drug being permeable so it could be beneficial to treat AD, and other diseases that show synucleinopathies. Although it is a small study, it is shown as a communication and its publication is proposed with some minor corrections:

First: Make and include a figure explaining the effect of the fludil in the introduction.

Second: Add figure captions throughout the text

Third: Redo table 1 and 2, so that the numbers and letters are displayed correctly

 Minor editing of English language required

Author Response

The manuscript titled “Neurodegenerative disease associated pathways in the brains of triple transgenic Alzheimer’s model mice are reversed following two weeks peripheral administration of fasudil”, shows an analysis of the transcriptional effects of peripherally administered fasudil in brain tissue, the drug being permeable so it could be beneficial to treat AD, and other diseases that show synucleinopathies. Although it is a small study, it is shown as a communication and its publication is proposed with some minor corrections:

First: Make and include a figure explaining the effect of the fludil in the introduction.

Second: Add figure captions throughout the text

Third: Redo table 1 and 2, so that the numbers and letters are displayed correctly

Unfortunately, the MS got somewhat corrupted upon pdf generation. We have now loaded a pdf version of the paper. Hopefully, this will now clarify our work.